# Comprehensive analysis of *cis-* and *trans-*acting factors affecting ectopic Break-Induced Replication

**Tannia Uribe-Calvillo**[1], **Laetitia Maestroni**[1], **Marie-Claude Marsolier**[2,3], **Basheer Khadaroo**[1], **Christine Arbiol**[1¤], **Jonathan Schott**[1], **Bertrand Llorente**[1]*

1 Cancer Research Center of Marseille, CNRS UMR7258, Inserm U1068, Institut Paoli-Calmettes, Aix-Marseille Université UM105, Marseille, France, 2 Institute for Integrative Biology of the Cell (I2BC), Institut des sciences du vivant Frédéric Joliot, CNRS UMR 9198, CEA Saclay, Gif-sur-Yvette, France, 3 Eco-anthropologie (EA), Muséum national d'Histoire naturelle, CNRS, Université de Paris, Musée de l'Homme, Paris, France

¤ Current address: Toulouse Biotechnology Institute, Université de Toulouse, CNRS, INRA, INSA, Toulouse, France

* bertrand.llorente@inserm.fr

**Data Availability Statement:** All relevant data are within the manuscript and its Supporting information files.

## Abstract

Break-induced replication (BIR) is a highly mutagenic eukaryotic homologous DNA recombination pathway that repairs one-ended DNA double strand breaks such as broken DNA replication forks and eroded telomeres. While searching for cis-acting factors regulating ectopic BIR efficiency, we found that ectopic BIR efficiency is the highest close to chromosome ends. The variations of ectopic BIR efficiency as a function of the length of DNA to replicate can be described as a combination of two decreasing exponential functions, a property in line with repeated cycles of strand invasion, elongation and dissociation that characterize BIR. Interestingly, the apparent processivity of ectopic BIR depends on the length of DNA already synthesized. Ectopic BIR is more susceptible to disruption during the synthesis of the first ~35–40 kb of DNA than later, notably when the template chromatid is being transcribed or heterochromatic. Finally, we show that the Srs2 helicase promotes ectopic BIR from both telomere proximal and telomere distal regions in diploid cells but only from telomere proximal sites in haploid cells. Altogether, we bring new light on the factors impacting a last resort DNA repair pathway.

## Author summary

DNA is a long molecule composed of two anti-parallel strands that can undergo breaks that need to be efficiently repaired to ensure genomic stability, hence preventing genetic diseases such as cancer. Homologous recombination is a major DNA repair pathway that copies DNA from intact homologous templates to seal DNA double strand breaks. Short DNA repair tracts are favored when homologous sequences for the two extremities of the broken molecule are present. However, when homologous sequences are present for only one extremity of the broken molecule, DNA repair synthesis can proceed up to the end of

**Funding:** TU was supported by a PhD fellowship from CONACYT, Consejo Nacional de Ciencia y Tecnología, Mexico. BL team was supported by the Agence Nationale de la Recherche (ANR) grants ANR-13-BSV6-0012-01 and ANR-18-CE12-0013-01, and the Fondation ARC project SFI20121205448. The funders had no role in study design, data collection and analysis, decision to publish, or preparation of the manuscript.

**Competing interests:** The authors have declared that no competing interests exist.

the chromosome, the telomere. This notably occurs at eroded telomeres when telomerase, the enzyme normally responsible for telomere elongation, is inactive, and at broken DNA replication intermediates. However, this Break-Induced Replication or BIR pathway is highly mutagenic. By initiating BIR at various distances from the telomere, we found that the length of DNA to synthesize significantly reduces BIR efficiency. Interestingly, our findings support two DNA synthesis phases, the first one being much less processive than the second one. Ultimately, this tends to restrain the use of this last resort DNA repair pathway to chromosome extremities notably when it takes place between non-allelic homologous sequences.

## Introduction

Break induced replication (BIR) is a eukaryotic one-ended homologous DNA recombination (HR) process [1,2]. It is thought to deal with one-ended DNA double strand breaks (DSBs) such as those generated by the encounter of replication forks with single strand DNA (ssDNA) breaks, as well as other one-ended DSBs like eroded telomeres that can be repaired also outside S phase. Notably, a BIR-like mechanism is responsible for the alternative lengthening of telomeres (ALT) in the absence of telomerase in 10–15% of all cancers [3]. A similar mechanism is responsible for the emergence of survivors in the absence of telomerase in *Saccharomyces cerevisiae* [4]. BIR can therefore take place between homologous loci located at allelic positions on sister chromatids or non-sister chromatids, as well as at non-allelic positions. The first steps of BIR are common to any canonical HR reaction, up to the generation of the D-loop and the initiation of DNA repair synthesis from the 3' invading end by DNA polymerase delta. Two-ended DSB repair by HR requires generally rather short tracts of DNA repair synthesis that initiate by DNA polymerase delta from both 3' ends of the DSB after they anneal to a complementary template [5–7]. In the case of BIR, the unique 3' invading end allows DNA polymerase delta to prime DNA repair synthesis [8–10]. BIR associated DNA replication proceeds by migrating this D-loop potentially up to the telomere. This results in conservative DNA replication where the two newly synthesized strands are on the same chromatid [11,12]. Pif1 is the helicase responsible for branch migrating the D-loop over tens to hundreds of kilobases [13–15]. After about 30 kb of DNA synthesis by DNA polymerase delta, DNA polymerases alpha and epsilon come into play. This likely stabilizes the nascent DNA strand and promotes synthesis of the second strand of the BIR product, a process still not well understood during BIR [9]. While dispensable for DNA synthesis during S phase, for two-ended DSB repair and repair of short gaps, the Pol32 subunit of DNA polymerase delta is essential for BIR and for the repair of long gaps, likely promoting the processivity of this enzyme [10,16]. Overall, BIR associated DNA synthesis deviates from classical DNA repair synthesis as well as from S phase DNA synthesis both mechanistically and through the proteins involved.

BIR associated DNA synthesis is highly mutagenic, with high rates of point mutations, frameshifts and gross chromosomal rearrangements. Point mutations are mainly the result of the long ssDNA region generated behind the migrating D-loop ensuring DNA replication during BIR. These ssDNA regions can suffer base damage which promotes subsequent mutagenesis when replicated by the translesion DNA polymerase zeta [17]. The replication protein A (RPA) is important to protect these long ssDNA regions and ensure accurate BIR [18]. Interestingly, in addition to its D-loop disrupting activity [19], the Srs2 helicase is important during BIR to prevent Rad51 from mediating toxic interactions involving these long ssDNA regions [20]. DNA polymerase delta was shown to generate a high level of unrepaired frameshift

mutations during BIR [21]. Gross chromosomal rearrangements can form in the early phase of the BIR reaction where frequent template switching events were observed both between homologous chromosomes and between ectopic repeats [22]. This observation led to the suggestion that the early phase of BIR undergoes more frequent cycles of strand invasion, elongation and dissociation than the late phase of BIR, but the mechanism remains unknown [17,22]. Gross chromosomal rearrangements can also occur when the BIR template is damaged or when the DNA synthesis step is hampered such as in the absence of Pol32 or in the Pol3-ct mutant [23–25]. Notably, BIR intermediates are destabilized by interstitial telomeric repeats which can abort BIR reactions [9,26]. This leads to stable aborted BIR products thanks to the action of telomerase on the dissociated end.

Several other factors related to the structure of the template to replicate also inhibit BIR. The BIR reaction is blocked by a converging replication fork when initiated from a one-ended DSB generated during S phase by the passage of a replication fork through an inducible ssDNA break [27]. The BIR reaction is also blocked by RNA-DNA hybrids normally resolved by RNases H; by RNA polymerase I transcription at the rDNA locus [28], and eventually by converging RNA polymerase II transcription close to the BIR initiating point [9]. In addition, BIR is impaired when the template to replicate is heterochromatic [29]. Last but not least, BIR efficiency, defined as repair efficiency by BIR reactions, seems to be inversely correlated to the length of DNA to replicate, but the nature of such a relationship is still unclear [10,11,30].

Here, while searching for cis-acting factors regulating ectopic BIR, we notably found that the variations of ectopic BIR efficiency as a function of the length of DNA to replicate can be modelled as a combination of decreasing exponential functions, in agreement with a mechanism involving repeated cycles of strand invasion, extension and dissociation. More precisely, our data support two phases during ectopic BIR, a first phase with low apparent DNA synthesis processivity followed by a phase with higher apparent DNA synthesis processivity, likely determined by the DNA polymerases involved [9]. Consequently, ectopic BIR initiated close to chromosome ends is much more efficient than ectopic BIR initiated at a more distal position, and we bring a definitive demonstration of this using CRISPR-Cas9-induced reciprocal translocations to modulate the length of DNA to synthesize from a unique ectopic BIR initiating locus. This ectopic BIR property, combined with all the other factors impeding BIR, participates to make BIR a last resort DSB repair mechanism.

## Results

### Ectopic BIR efficiency is higher close to chromosome ends

In order to look for cis-acting factors regulating ectopic BIR efficiency, we initiated BIR along the one megabase-long right arm of chromosome IV, the longest *S. cerevisiae* chromosome arm apart from the right arm of chromosome XII that contains the rDNA locus. We took advantage of the yeast deletion library in which the *KANMX4* cassette is the selection marker that replaces every ORF [31], in order to use this cassette as a unique homologous region to initiate BIR with the same chromosome fragmentation vector from virtually any genomic locus (Fig 1) [32–34]. More precisely, the transcription orientation of the *KANMX4* cassette being the same as the deleted ORFs, we targeted BIR only at positions where the transcription orientation of the *KANMX4* cassette is directed toward the telomere using a single chromosome fragmentation vector [34]. We expressed BIR efficiency as the number of transformants with the linearized chromosome fragmentation vector initiating BIR from *KANMX4* normalized by the number of transformants with the linearized chromosome fragmentation vector initiating BIR from the control *BUD3* locus located ca. 100 kb away from the left telomere of chromosome III [32,33]. We initiated BIR from 11 loci spread roughly every 100 kb between

A.

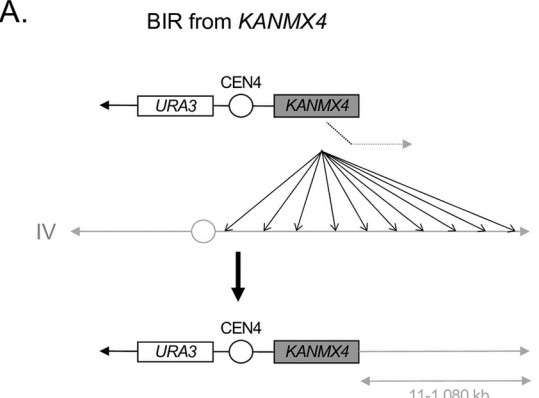

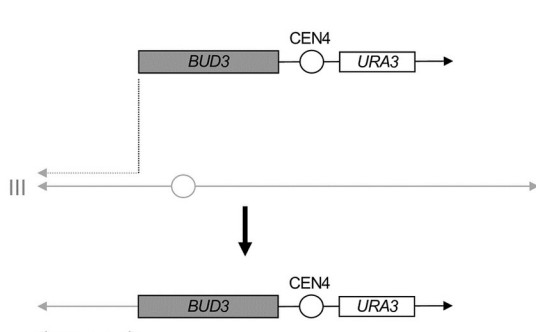

B.

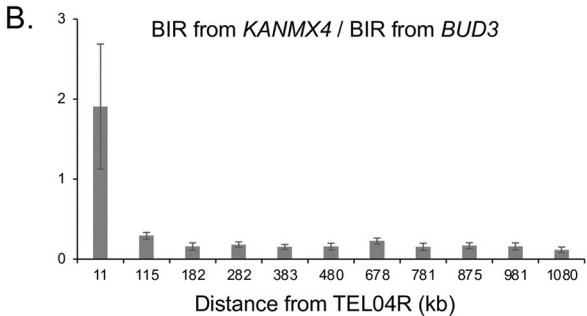

**Fig 1. Ectopic BIR efficiency along the right arm of chromosome IV. A**. The BIR assay used in this study is based on the transformation into yeast cells of a linearized chromosome fragmentation vector (CFV). This CFV contains a telomere seed (black arrowhead), the *URA3* auxotrophic marker, the centromere from chromosome IV (CEN4) and a region of homology with the genome to initiate BIR (*KANMX4* or *BUD3*). This CFV being deprived of any origin of replication, the only way it can be maintained into the cell is after undergoing BIR to incorporate an origin of replication from the chromosome as well as a telomere to be stabilized. The number of transformants obtained using the linearized plasmid reflects BIR efficiency. **Left**: the region of homology with the genome is the *KANMX4* selection cassette. This cassette has been used to systematically replace individually all the *S. cerevisiae* ORFs keeping the same orientation of transcription as the endogenous ORFs [31]. **Right**: the region of homology with the genome encompasses *BUD3*. **B**. Ratio between the efficiencies of BIR initiated from the *KANMX4* selection cassette from different loci along the right arm of chromosome IV and the efficiency of BIR initiated from *BUD3*. Mean ratios of five (1,080; 981; 875; 11), four (781; 678; 383; 282; 182) and three (480; 115) independent experiments are shown. Error bars represent standard deviation.

the centromere and the telomere of chromosome IV (TEL04R), the closest locus to the telomere being *YDR541c* whose 3' end is located ca. 11 kb away from TEL04R. BIR efficiency was roughly constant for all loci except for the *YDR541c* locus for which it was significantly higher (Fig 1B). This suggests that telomere proximity behaves as a cis-acting factor facilitating ectopic BIR.

## Ectopic BIR efficiency from a chromosome fragmentation vector fits two exponential functions of the DNA length to synthesize

The efficiency of BIR initiated at the *KANMX4* locus located 115 kb away from TEL04R is lower than the efficiency of BIR initiated at the *BUD3* locus located at a similar distance from the left telomere of chromosome III (Fig 1B). This suggests that BIR is less efficient when using *KANMX4* as a homology region than when using *BUD3*. This likely results from the longer homology region of the *BUD3* cassette compared to the *KANMX4* cassette (ca. 5 kb and 1.5 kb, respectively) as previously observed [10].

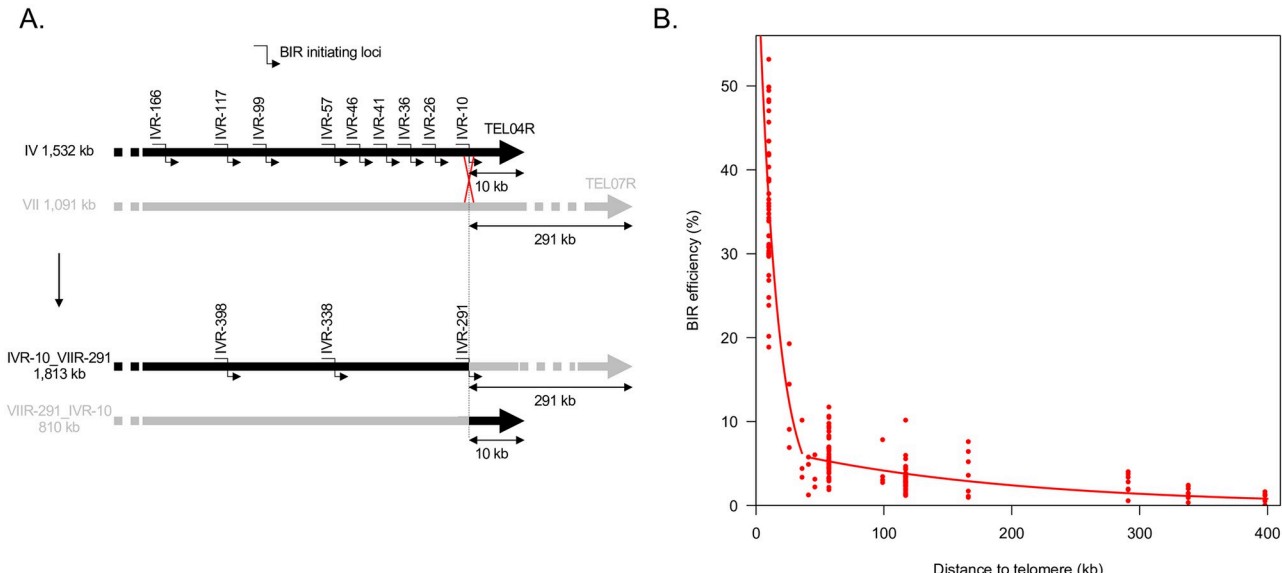

**Fig 2. A**. Diagram representing the twelve BIR initiating loci used to generate Fig 2B (S3 Table). Top: wild type strains with nine BIR initiating loci on the right arm of chromosome IV. Bottom: translocated strains with three BIR initiating loci IVR-398, IVR-338 and IVR-291 corresponding to the IVR-117, IVR-57 and IVR-10, respectively, from the parental non-translocated strains. **B**. BIR efficiency modelled as a combination of two exponential functions of the length of DNA to synthesize. BIR efficiency is expressed as the ratio between the number of transformants from the linearized chromosome fragmentation vector and the number of transformants from the circular chromosome fragmentation vector containing an autonomous replicating sequence (ARS). The two red curves represent exponential law fits. See main text for details.

To optimize the analysis of BIR efficiency as a function of the distance to the telomere, we switched to ca. 5 kb homology regions cloned into the chromosome fragmentation vector instead of *KANMX4*. In addition to the regions encompassing *YDR541c* and *YDR479c* corresponding to the previous loci located 11 and 115 kb away from the telomere, respectively, we analyzed ten other loci (Fig 2A and S3 Table). Seven of these loci are located between 166 and 26 kb from TEL04R. In order to test longer distances away from the telomere without involving additional BIR initiating loci which could have increased the probability of including local effects on BIR efficiency, we generated a reciprocal translocation that exchanged the 10 kb of chromosome IV proximal to TEL04R with the 291 kb of chromosome VII proximal to TEL07R. We performed the chromosome translocation as described by Fleiss and Fischer using CRISPR-Cas9-induced DSBs and oligonucleotides overlapping the translocation breakpoints as templates [35]. The loci originally located 10, 57 and 117 kb from TEL04R are located 291, 338 and 398 kb from the right telomere in this translocated strain. For simplicity, we refer to the BIR initiating loci by their chromosome arms and the distance in kb to the telomere. Overall, these loci range from IVR-10 to IVR-398 (Fig 2A).

We measured BIR efficiency for these twelve loci as the ratio between the number of transformants with the linearized chromosome fragmentation vector initiating BIR, and the number of transformants with a circular vector (Fig 2). Overall, we observed that BIR efficiency decreases with the size of DNA to synthesize, but the relationship is not linear. Since BIR is likely characterized by repeated cycles of strand invasion, elongation and dissociation [22], we thought that BIR efficiency, hereafter designated *E(dist)* as a function of the DNA length to synthesize (*dist*), should be best described by exponential functions. The data were poorly fitted by a single exponential function (value of Akaike Information Criterion [AIC] equal to -584.2), so we tested combinations of two exponential functions describing separately the results for the first *x* and for the last 12-*x* distances (in increasing order) with *x* varying

between 2 and 10. Similar fits with low, close AIC values were observed for combinations of two exponential functions when the first function fitted the data either from IVR-10 to IVR-26, or from IVR-10 to IVR-36, or from IVR-10 to IVR-41, or from IVR-10 to IVR-46 (AIC values equal to -622.5, -622.5, -623.0, and -623.6, respectively, see S1 Fig). Fig 2B shows the curves corresponding to the first exponential function fitting the data from IVR-10 to IVR-36; this combination of exponential curves presents the additional advantage that the ends of the two curves meet without any shift.

The good fit of the data with a combination of two exponential functions can be interpreted by the following model involving two successive DNA replication modes during BIR. The BIR efficiency, *E(dist)*, represents the probability that a DSB end with homology to a chromosome region located at *dist* kb away from the telomere is properly repaired by BIR (thus ensuring cell viability), that is the probability that BIR proceeds as far as the distance *dist*. Let *k* be the probability that BIR is initiated at the DSB end, and let's consider first short distances (the first exponential curve). If we assume that DNA synthesis for these short DNA lengths goes on with a constant probability $p_1$ for each kb (which means that it has a probability *$1-p_1$* to be irreversibly disrupted at each kb), the probability that *dist* kb are replicated in a row is $p_1^{dist}$ (which corresponds to the probability $p_1$ that the first kb is replicated, multiplied by the same probability $p_1$ that the second kb is replicated, etc. up to the last kb). *E(dist)* is thus equal to the probability of initiation *k* multiplied by $p_1^{dist}$, hence the equation of the first fitting exponential curve:

$$E(dist) = kp_1^{dist}.$$

Replication characteristics change after the synthesis of a certain DNA length. The value of this threshold, *T*, can be estimated at about 35–40 kb, at the point where the two fitting curves meet (Fig 2B and S1 Fig). After this threshold, DNA synthesis goes on with a higher, constant probability $p_2$ to proceed for each kb, and we have, for *dist > T*, the equation:

$$E(dist) = kp_1^T p_2^{(dist-T)},$$

which corresponds to the second exponential fitting curve.

The fitting curves shown in Fig 2B correspond to values of *k*, $p_1$ and $p_2$ equal to 0.71 +/- 0.07, 0.935 +/- 0.009, and 0.995 +/- 0.002, respectively. According to our model, this means (i) that BIR has a probability of 0.71 to be initiated for cells transformed with the linearized chromosome fragmentation vector (if we assume that transformation efficiencies are identical for the circular vector and for the linearized chromosome fragmentation vectors, and that BIR is the only limiting factor for the maintenance of this artificial chromosome), and (ii) that over the first 35–40 kb the probability that BIR will be irreversibly disrupted is 0.065 for each kb (1-$p_1$), after which the probability falls by a factor of 13, to 0.005 (1-$p_2$).

Importantly, BIR events from telomere proximal and telomere distal regions strongly depend on *POL32*, supporting long tract DNA repair synthesis and therefore *bona fide* BIR from all studied loci (S2 Fig). The apparent weaker dependency on *PIF1* likely comes from the use of the hypomorphic allele *pif1m2* that still partially localizes to the nucleus (S2 Fig). *Bona fide* BIR is further supported by PFGE analysis of clones transformed with the chromosome fragmentation vectors that revealed chromosome fragments at the expected size for BIR products (S3 Fig).

## Direct evidence that ectopic BIR efficiency depends on the size of DNA to synthesize

In order to directly demonstrate that the length of DNA to synthesize determines ectopic BIR efficiency independently of the initiation locus, we performed targeted chromosome

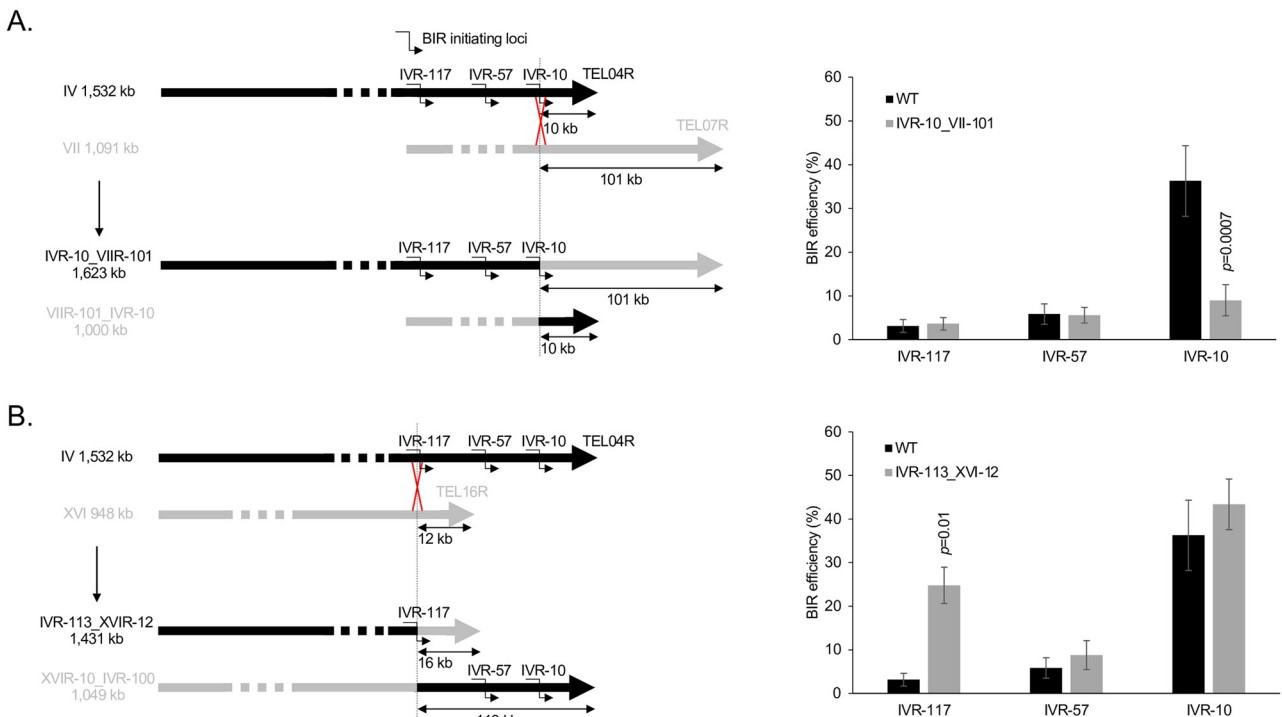

**Fig 3. Ectopic BIR efficiency in translocated strains where the distance between the BIR initiating loci and the telomere was modified. A**. The translocation exchanges the last 101 kb of the right arm of chromosome VII with the last 10 kb of the right arm of chromosome IV, increasing by 91 kb the length of DNA to synthesize for BIR initiated at the IVR-117, IVR-57 and IVR-10 loci. WT: reference (wild type) strain. **B**. The translocation exchanges the last 12 kb of the right arm of chromosome XVI with the last 113 kb of the right arm of chromosome IV. This decreases by 101 kb the length of DNA to synthesize for BIR initiated at the IVR-117 locus, but does not modify the distance to the telomere of the IVR-57 and IVR-10 loci. The WT values are the same in the two graphs and represent the means of 49 independent experiments. Values for the translocated strains represent the means of three independent experiments. Error bars represent standard deviations. Indicated are Student's *t*-test *p*-values <0.05 when comparing a given point from a translocated strain with its equivalent from the parental strain without translocation.

translocations to modify the distance between chromosome IV BIR initiating loci and the telomere (Fig 3). We generated the IVR-10_VIIR-101 translocated strain where the 101 kb telomeric fragment from the right arm of chromosome VII replaces the 10 kb telomeric fragment from the right arm of chromosome IV. In this strain, the length of DNA to synthesize for BIR events initiated upstream of the translocation point on chromosome IV increases by 91 kb (Fig 3A). BIR efficiency from the most telomere proximal IVR-10 locus shows a clear drop in this translocated strain compared to the parental strain. This latter locus shows a BIR efficiency comparable to the BIR efficiency measured from the IVR-117 locus in the parental strain (Figs 2 and 3A). We also generated the IVR-113_XVIR-12 translocated strain where the 12 kb telomeric fragment from the right arm of chromosome XVI replaces the 113 kb telomeric fragment from the right arm of chromosome IV (Fig 3B). In this strain, the length of DNA to synthesize for BIR events initiated upstream of the translocation point on chromosome IV decreases by 101 kb, positioning the IVR-117 locus 16 kb away from the telomere. BIR efficiency from this locus increases in the translocated strain to 25 +/- 4% compared to 3 +/- 1% in the parental strain. In this translocated strain, the distance to the telomere of the IVR-57 and IVR-10 loci is unchanged and the corresponding BIR efficiencies are similar between the translocated and the parental strain (Fig 3B).

Overall, these results show that the length of DNA to synthesize during ectopic BIR directly influences BIR efficiency independently of the initiating locus.

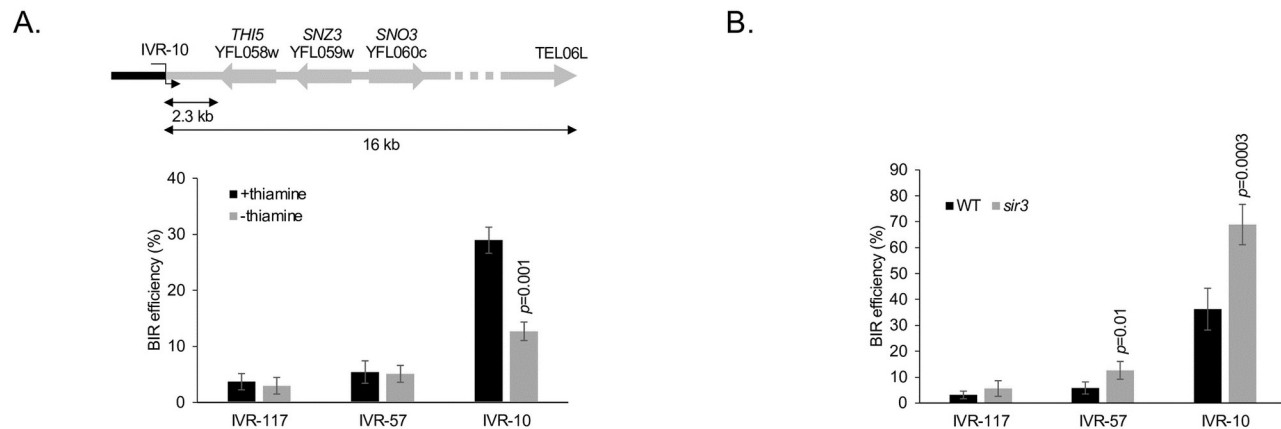

**Fig 4. Ectopic BIR efficiency is decreased by cis-acting factors including transcription and heterochromatin of the template chromatid. A**. A chromosome translocation positioned the last 16 kb of the left arm of chromosome VI downstream of the IVR-10 BIR initiating locus. This region contains the *THI5*, *SNZ3* and *SNO3* genes whose transcription is active in the absence of thiamine in the growth medium and inactive in its presence. The arrows indicate the transcription orientation of the corresponding genes. Values are the means of three independent experiments in the presence of thiamine and four experiments in the absence of thiamine. Indicated is the Student's *t*-test *p*-value <0.05 when comparing BIR efficiency with or without thiamine in the growth medium. **B**. BIR efficiency from the IVR-117, IVR-57 and IVR-10 loci in a *sir3* null mutant. WT values are identical to those in Fig 3. The *sir3* values correspond to the means of five independent experiments. Indicated are Student's *t*-test *p*-values <0.05 when comparing BIR efficiency from a given locus in the mutant strain with the BIR efficiency from the same locus in the WT strain. Error bars represent standard deviations.

## Transcription and heterochromatin of the template impair ectopic BIR efficiency

In addition to the length of DNA to synthesize, other cis-acting chromosomal parameters may affect ectopic BIR efficiency. Both RNA polymerase I and II mediated transcription were shown to impair BIR [9,28]. We therefore tested the effect of transcription on BIR in our assay system by comparing BIR efficiency from the same locus with and without downstream transcription. The thiamine regulon composed of *THI5*, *SNO3* and *SNZ3* located in the left subtelomere of chromosome VI is transcribed in the absence of thiamine in the medium and switched off in the presence of thiamine [36] (S4A Fig). We performed a reciprocal translocation to position this regulon just downstream of the IVR-10 locus (Fig 4A). In the translocated strain, the IVR-10 locus is positioned about 16 kb away from the telomere. BIR efficiency from this locus in the translocated strain is 29 +/- 2% in the presence of thiamine compared to 13 +/- 2% in the absence of thiamine (Fig 4A). We conclude that transcription downstream of a BIR initiating locus impairs ectopic BIR initiated from a chromosome fragmentation vector.

The Silent Information Regulator Sir proteins promote heterochromatinization of subtelomeres. We looked at the effect of *SIR3* deletion which encodes a structural component of the Sir complex. We found a significant increase in BIR efficiency from the IVR-10 and IVR-57 loci in the absence of *SIR3* (Fig 4B). This is compatible with heterochromatin impairing ectopic BIR when located downstream and close to the BIR initiation locus only, but not when BIR events are initiated farther away.

## Effect of trans-acting factors on ectopic BIR

In addition to cis-acting factors, other trans-acting factors may affect ectopic BIR and contribute to the low efficiency of BIR when it initiates far away from the telomere. A *DUN1*-dependent increase in the level of dNTPs is observed during BIR occurring in G2/M [21] as well as during the formation of type II survivors in the absence of telomerase [37]. The fact that BIR

A.

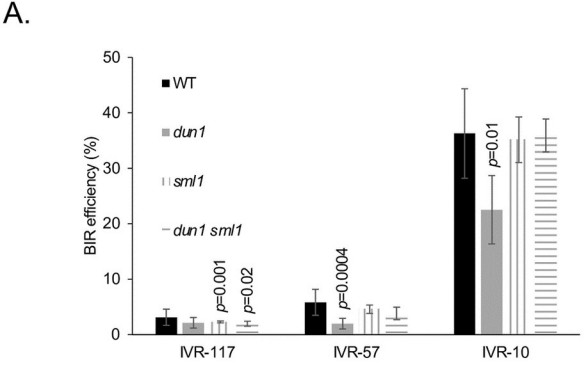

C.

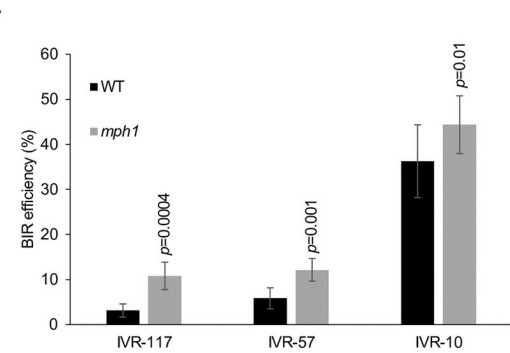

B.

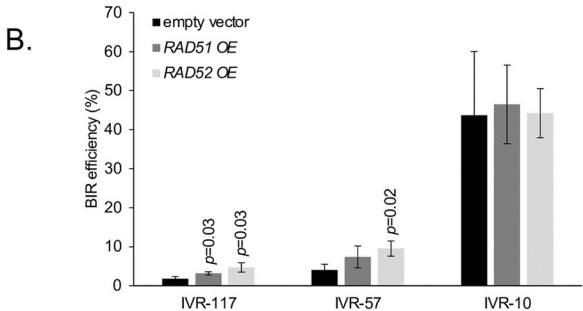

**Fig 5. Ectopic BIR efficiency is affected by the trans-acting factors Dun1, Rad51, Rad52 and Mph1.** Indicated are Student's *t*-test *p*-values <0.05 when comparing BIR efficiency from a given locus in a mutant strain with the BIR efficiency from the same locus in the WT strain. Error bars represent standard deviations. **A**. BIR efficiency of the null mutants *dun1*, *sml1* and *dun1 sml1* from the IVR-117, IVR-57 and IVR-10 loci. WT values are identical to those in Fig 3. Mutant values are the means of four independent experiments for *dun1* and three independent experiments for *sml1* and *dun1 sml1*. **B**. Similar to A with *RAD51* and *RAD52* overexpression (OE) from a multicopy vector. Values are the means of three independent experiments for each condition. **C**. Similar to A with the *mph1* null mutant. WT values are identical to those in Fig 3. Values for the *mph1* strain are the means of seven independent experiments for IVR-117 and IVR-10 and six independent experiments for IVR-57.

requiring long DNA synthesis tracts is less efficient than BIR requiring short DNA synthesis tracts could indicate that the dNTP pool is a limiting factor for BIR in the former situation. We tested this possibility by deleting the ribonucleotide reductase inhibitor Sml1 which is known to increase the dNTP pool [21]. This did not increase BIR efficiency, but instead was associated with a slight decrease in BIR efficiency from the IVR-117 locus only (Fig 5A). On the contrary, in the absence of Dun1, which has many functions in response to DNA damage including Sml1 degradation and therefore up-regulation of the ribonucleotide reductase activity, we observed a reduction in BIR efficiency from IVR-57 and IVR-10 loci (Fig 5A). Despite this decrease, BIR efficiencies from IVR-10 is still much higher than BIR efficiency from IVR-57 in a wild type context, once again underlining the major contribution of telomere proximity to ectopic BIR efficiency.

Overexpressions of Rad51 and Rad52 were shown to increase BIR efficiency using a chromosome based assay [18,38]. In our plasmid-based assay, overexpression of Rad52 increases BIR efficiency from the IVR-117 and IVR-57 loci while the increase is significant only from IVR-117 when Rad51 is overexpressed (Fig 5B). The absence of effect of Rad51 and Rad52 overexpression on BIR initiated at the IVR-10 locus may come from shorter ssDNA intermediates during this BIR reaction that would not require extra Rad51 and Rad52 to be stabilized. Overall, BIR efficiency under these overexpression conditions is still low, and BIR initiated close to TEL04R from the IVR-10 locus still much higher than BIR efficiency from the IVR-57

and IVR-117 loci when Rad52 or Rad51 are over-expressed. In contrast to Rad51 and Rad52, the helicase Mph1 counteracts BIR [29,39,40]. Consistent with this, in the absence of Mph1 we observed increased BIR efficiency from the IVR-117, IVR-57 and IVR-10 loci (Fig 5C).

### Srs2 promotes ectopic BIR both at telomere proximal and distal sites in diploid cells, but only close to telomeres in haploid cells independently of Y' repeated elements

Srs2 is crucial to prevent Rad51 mediated toxic interactions involving the long ssDNA tract exposed during BIR [20]. Therefore, Srs2 might be more important for BIR reactions initiated from telomere distal loci compared to telomere proximal loci because they may expose longer ssDNA tracts [17]. To our surprise, in a haploid background, the absence of Srs2 decreases BIR efficiency only from the IVR-10 locus (Fig 6A). On the contrary, in a diploid background, BIR efficiency is decreased from all loci in the absence of Srs2 (Fig 6D). Because some Srs2 functions are affected by heterozygosity at the *MAT* locus but not by the ploidy itself [41], we deleted the *MATa* locus from the previous diploid strain. In the absence of Srs2, BIR

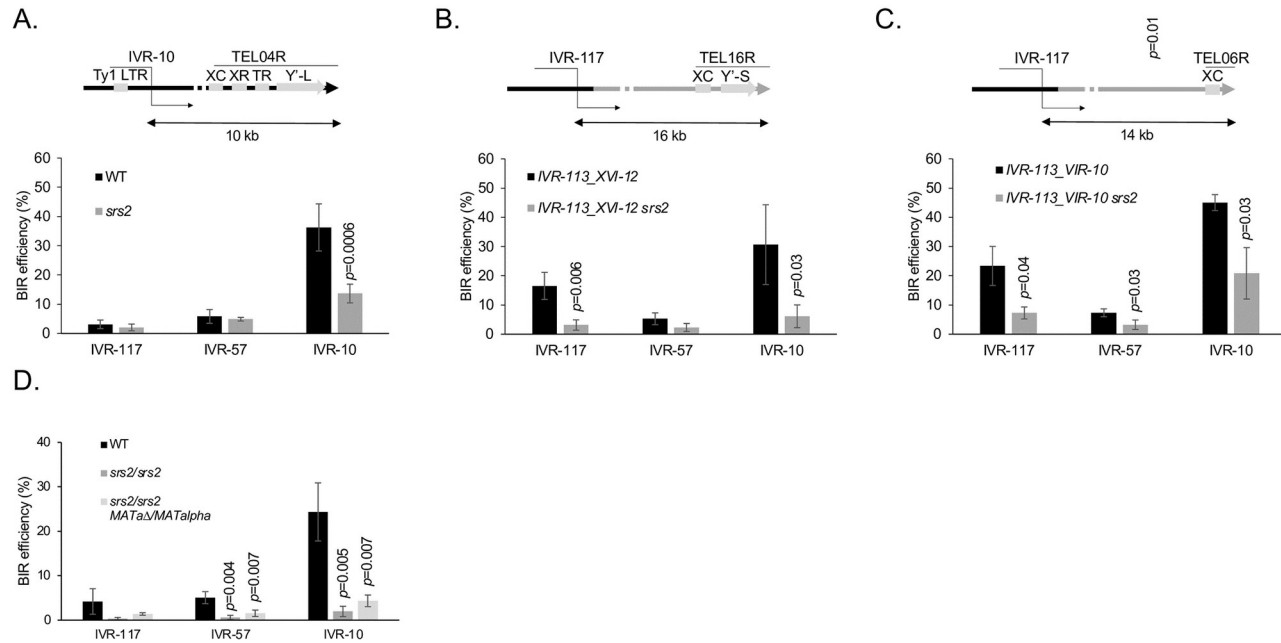

**Fig 6. Ectopic BIR efficiency in the absence of Srs2. A**. Parental strain configuration. The IVR-10 locus overlaps a Ty1 LTR (YDRWdelta31). The telomeric region of the right arm of chromosome IV TEL04R is composed of an X element core sequence (XC), a ca. 475 bp sequence common to all chromosome ends; X element combinatorial repeats (XR); an internal stretch of telomeric repeats (TR); a long Y' element (Y'-L) and a terminal stretch of telomeric repeats (right arrowhead). In the absence of Srs2, BIR efficiency decreases from the IVR-10 locus only. **B**. The last 113 kb of the right arm of chromosome IV were replaced by reciprocal translocation with the last 12 kb of the right arm of chromosome XVI. The telomeric region of the right arm of chromosome XVI TEL16R is composed of an XC; a short Y' element (Y'-S) and a terminal stretch of telomeric repeats. The IVR-117 locus is 16 kb away from the telomere in the translocated strain and does not overlap any LTR. In the absence of Srs2, BIR efficiency decreases from the IVR-117 and IVR-10 loci that are located 16 and 10 kb away from the flanking telomeres TEL16R and TEL04R, respectively. **C**. The last 113 kb of the right arm of chromosome IV were replaced by reciprocal translocation with the last 10 kb of the right arm of chromosome VI. The telomeric region of the right arm of chromosome VI TEL06R is composed of an XC and a terminal stretch of telomeric repeats. The IVR-117 locus is 14 kb away from the telomere in the translocated strain and does not overlap any LTR. In the absence of Srs2, BIR efficiency decreases from the IVR-117 and IVR-10 loci that are located 14 and 10 kb away from the flanking telomeres TEL06R and TEL04R, respectively. **D**. Effect of ploidy and mating type heterozygosity on BIR efficiency. WT values in panel A are the same as in Fig 3. The *srs2* values of panel A and the values of panel C are the means of three independent experiments. The values of panels B and D are the means of four experiments. Indicated are Student's *t*-test *p*-values <0.05 when comparing BIR efficiency from a given locus in a mutant strain with the BIR efficiency from the same locus in the relevant reference strain. Error bars represent standard deviations.

efficiencies were decreased in a diploid background with and without *MATa*, but slightly less without *MATa* (Fig 6D). This shows that the BIR defect of a *srs2* null mutant in a diploid background is mainly due to the ploidy.

The decrease of BIR efficiency from the telomere proximal locus in the haploid background in the absence of Srs2 may result from the presence of surrounding repeated sequences that may interfere with BIR. In the case of the right end of chromosome IV, these repeated sequences include the solo delta LTR YDRWdelta031 present in the IVR-10 locus that is used to initiate BIR from the fragmentation vector. In addition, from left to right the repeats from TEL04R comprise an X element core sequence, X element combinatorial repeats, a short stretch of telomeric repeats, and a long Y' element before the telomeric repeats (Fig 6A). To distinguish between the effect of the solo LTR and the subtelomeric repeats in the absence of Srs2, we took advantage of the translocated strain where the last right 113 kb of chromosome IV are replaced by the last right 12 kb of chromosome XVI (Figs 3B and 6B). In this translocated strain, the IVR-117 locus initially positioned 117 kb away from the telomere is now positioned 16 kb away from it. Importantly, this BIR initiating region is devoid of LTR, and the last right 12 kb of chromosome XVI also contain a X element core sequence and a short Y' element in addition to the telomeric repeats. In the presence of Srs2 in this translocated strain, BIR initiated from the IVR-117 locus is about five times more efficient than in the absence of Srs2. This observation phenocopies what is observed in the parental strain and in the IVR-113_XVI-12 background itself at the most telomere proximal locus IVR-10. These results show that in the absence of Srs2 the defect in BIR efficiency at the most telomere proximal loci is independent of the LTR present in the BIR initiating region of chromosome IV. This BIR defect may therefore come from other repeats common to TEL04R and TEL16R, which include the X element core sequence, Y' elements and telomeric repeats.

The X element core sequence and the telomeric repeats are common to all telomeres but not the Y' elements. To test the effect of the presence of Y' elements on BIR, we translocated the last 10 kb of the right end of chromosome VI comprising an X element core sequence and telomeric repeats only to replace the last 113 kb of the right arm of chromosome IV (Fig 6C). In the absence of Srs2, this translocated strain showed a BIR defect when it was initiated at the two telomere proximal loci IVR-117 and IVR-10. This shows that the effect of Srs2 on BIR initiated near telomeres is independent of Y' elements. Whether such an effect depends on the X element core sequence and / or the telomeric repeats remains to be determined.

## Srs2 restrains ectopic recombination

The YDRWdelta031 LTR from the IVR-10 BIR initiating locus does not impact BIR efficiency in the absence of Srs2 since BIR efficiency is not significantly different for the IVR-117 and the IVR-10 initiating sites neither in the IVR-113_XVI-12 *srs2* strain nor in the IVR-113_VIR-10 *srs2* strain (Fig 6B and 6C). However, this element may promote ectopic interactions. We analyzed by PFGE followed by Southern blot the sizes of the BIR products from the chromosome fragmentation assay in various strain backgrounds (S3 Fig). Initiating BIR at the IVR-10 locus, we observed 21% (15 out of 70 clones) of products of aberrant size in a wild type background (S3A Fig), which come from either ectopic BIR due to the YDRWdelta031 or from template switching during the BIR reaction involving other repeats [22,30]. The frequency of products of aberrant size dropped to 2% (2 out of 89 clones) when BIR was initiated from the IVR-117 locus positioned 16 kb upstream of TEL16R in the translocated IVR-113_XVI-12 strain (S3C Fig). This significant difference (*p*-value = 0.00012, Fisher's exact test) suggests that the presence of the LTR within the BIR initiating IVR-10 locus is responsible for most of the BIR products of aberrant size.

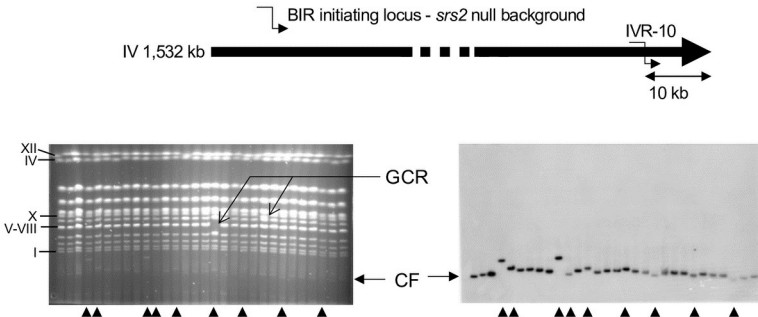

**Fig 7. PFGE and Southern blot of BIR products initiated at the IVR-10 locus in the absence of Srs2.** Ethidium bromide stained gels (left) and corresponding radioactive signal from Southern blot using a *URA3* probe specific of the chromosome fragmentation vector (right) are shown. Arrowheads indicate chromosome fragments (CF) of abnormal size resulting from BIR initiated by the chromosome fragmentation vector. Gross chromosomal rearrangements (GCR) are indicated by arrows. Roman numbers on the left correspond to *S. cerevisiae* chromosome numbers.

In addition, BIR initiated from the IVR-117 locus in the wild type background shows only 3% (3 out of 97 clones) of BIR products of abnormal size (S3B Fig). Since this BIR initiating locus and its surroundings are devoid of repeats, this significant difference (*p*-value = 0.00022, Fisher's exact test) with respect to the IVR-10 initiating locus suggests that BIR products of abnormal size are favored by repeated sequences at or in the downstream proximity of the BIR initiating locus. Indeed, downstream repeats far away from the BIR initiating locus such as the YDRWdelta031 LTR, as well as the other subtelomeric repeats including the long Y' elements from TEL04R do not promote as many detectable template switching events.

In the absence of Srs2, the frequency of BIR products of aberrant size from the IVR-10 locus (28 out of 100 clones, S3D Fig) is not statistically different from that observed in the presence of Srs2 (15 out of 70, S3A Fig) (*p*-value = 0.37; Fisher's exact test). This contrasts with the role of Srs2 in preventing template switching during recombination dependent replication at stalled replication forks in *Schizosaccharomyces pombe* [42]. However, the fraction of BIR products longer than expected is similar to the fraction of BIR products smaller than expected in the absence of Srs2 (15 versus 13, respectively, S3D Fig), while it is much lower in a WT background (2 versus 13, respectively, S3A Fig). This significant difference (*p*-value = 0.02; Fisher's exact test) suggests a qualitative difference in the mechanism generating BIR products of abnormal size in the absence of Srs2. In addition, among the 100 *srs2* clones resulting from BIR initiated from the IVR-10 locus, we observed four clones where at least one endogenous chromosome did not migrate at its expected size (Fig 7 and S3D Fig). Such a situation was never observed in the presence of Srs2 out of 256 clones analyzed, which corresponds to a significant difference with respect to *srs2* clones (*p*-value = 0.006, Fisher's exact test). These 256 clones include the 70 clones where BIR was initiated from the IVR-10 locus (S3A Fig), the 97 clones where BIR was initiated from the IVR-117 locus (S3B Fig) and the 89 clones where BIR was initiated from the IVR-117 locus in the IVR-113_XVI-12 translocated strain (S3C Fig). Furthermore, we did not observe any gross chromosomal rearrangement (GCR) out of 100 clones obtained by transforming the *srs2* null mutant with the circular chromosome fragmentation vector used as a control (S3E Fig). Although the numbers of GCR observed in *srs2* clones transformed with the linear chromosome fragmentation vector and the control vector are not significantly different (*p*-value = 0.1212, Fisher's exact test), these results suggest that the GCRs observed depend on the BIR reaction initiated in a *srs2* null background. These GCRs support the existence of recombination intermediates involving multiple chromosomes that have been resolved by structure specific nucleases [43]. Such intermediates can arise

through template switching during BIR or through multi-invasions [22,44], and are compatible with Srs2 undoing excessive Rad51-mediated ectopic DNA interactions during BIR in a wild type context.

## Discussion

### Increased efficiency of ectopic BIR when initiated close to chromosome ends

Previous work already revealed a high BIR efficiency close to chromosome ends during the repair of I-SceI induced DSBs [45]. Other reports showed an inverse correlation between the size of DNA to synthesize during BIR and BIR efficiency. Using a chromosome fragmentation assay similar to ours, Davis and Symington [33] observed a higher BIR efficiency at telomere proximal compared to telomere distal positions when BIR was initiated on chromosomes II and III. In addition, they observed maximum BIR efficiency when initiation was in a Y' element. Although they related this observation to the repeated nature of the Y' element, the present study shows that the proximity to the telomere is the major determinant of this elevated BIR efficiency. Using a chromosome assay where BIR is triggered by DSB induction at an HO site, Lydeard et al. [10] made observations compatible with an inverse correlation between the size of DNA to synthesize during BIR and ectopic BIR efficiency, which was further supported by Donnianni and Symington [11]. Using a more systematic approach monitoring BIR efficiency along the right arm of chromosome IV, we could establish that the inverse correlation between ectopic BIR efficiency and the size of DNA to synthesize during BIR is well fitted by a combination of two exponential functions. The quality of the fit by exponential functions of the size of DNA to replicate supports a model where BIR occurs by repeated cycles, in this case cycles of initiation, extension and dissociation.

From the fitting equations, we can notably extract the probability that DNA synthesis, once engaged, proceeds for another kb (as opposed to being irreversibly stopped). This parameter could be considered as an apparent processivity parameter of BIR-associated DNA synthesis. This processivity parameter increases after the synthesis of ~35–40 kb of DNA. This is reminiscent of the BIR reaction as proposed by Smith et al. [22] where the first part of the reaction is rather unstable with frequent template switches followed by a more stable part with fewer template switches. Our data suggest that the transition between these two stages of the BIR reaction takes place after the synthesis of ~35–40 kb DNA, while frequent template switching between homologs was restrained to the first 10 kb downstream of the BIR initiation locus in [22]. The reason for this apparent discrepancy is unclear. Interestingly, template switching during recombination dependent replication, a mechanism highly similar to BIR but induced by a replication block in *S. pombe* [46], was observed up to 75 kb from the replication blocking lesion. This could indicate that the distance over which the BIR replisome is highly unstable was originally underestimated [42]. Overall, it is tempting to relate the biphasic BIR behavior we observe with recent findings showing that BIR relies first on DNA polymerase delta for about 25–30 kb prior to involving DNA polymerases alpha and epsilon to complete the reaction [9]. The first unstable and consequently less processive phase of BIR would rely on DNA polymerase delta only. The second more stable and processive phase of BIR would rely on DNA polymerases delta, alpha and epsilon. The fact that Malkova and collaborators have observed similar, high BIR-associated mutation rates for a *ura3-29* base substitution reporter inserted at positions located 0 kb, 16 kb, and 90 kb away from the BIR initiation site suggests that both replication phases of BIR would be similarly mutagenic [17]. A major unanswered question is how the transition between these two phases occurs. So far, DNA polymerase alpha has been shown to interact with Rad51 at least in *Xenopus laevis* [47], with the

Cdc13-Stn1-Ten1 complex [48], and with the Mcm2-7-GINS-Cdc45 complex [49]. Whether any of these pathways is at play during long range BIR is unknown but deserves to be determined.

Interestingly, the limited dataset for BIR initiated at a chromosomal HO-induced DSB shows BIR efficiencies that are higher than those observed here for the chromosome fragmentation assay, and that decrease less sharply with increasing length of DNA to synthesize [11,18]. If transformation efficiencies are identical for the control vector and for the chromosome fragmentation vectors in our system, and if BIR is the only limiting factor for the repair of these vectors, these data suggest a higher probability of BIR initiation at the chromosomal HO-induced DSB. This could be due to a lower stability of the chromosome fragmentation vectors compared to a broken natural chromosome as a result of their sensitivity to the DNA end resection machinery [32]. The fact that BIR efficiency decreases more slowly with DNA length to synthesize for chromosomal HO-induced DSB than for the chromosome fragmentation vectors suggests a higher processivity of DNA synthesis in the former case.

Finally, it is intriguing that BIR between allelic loci from a disomic strain shows nearly 100% efficiency over a distance of 100 kb [9], while between ectopic loci, such as in our transformation-based assay and in chromosomal setups where BIR is initiated at an HO-induced DSB [10,11,33], BIR shows a lower efficiency and the length of DNA to replicate negatively impacts BIR efficiency. This suggests that allelic BIR benefits from an optimal configuration, that remains to be characterized, and that, once engaged, allows systematic completion of DNA synthesis up to the telomere while this is not the case for ectopic BIR.

## Cis- and trans-acting factors that inhibit ectopic BIR

**i. Transcription.**   We found that induction of transcription right downstream of a BIR initiating locus decreases BIR efficiency. The successive encounters with transcribed genes likely participate to the instability of the BIR associated DNA synthesis. Importantly, the transcription effect on BIR efficiency applies only for BIR events initiated at proximity of the transcribed region, but not 47 kb away from it. This suggests that only the first and less stable phase of BIR is sensitive to transcription from the template DNA. While we did not test it here, recent work showed that the transcription inhibitory effect on BIR is specific to converging transcription [9]. This may be particularly relevant when telomeres are maintained by BIR in the absence of telomerase. In this context, two classes of recombination can occur, among which the class I propagates Y' elements by BIR to all chromosome ends. Notably, for all the 17 out of 32 chromosome ends that contain at least one Y' element in the reference S288C strain, the transcription orientation of the Y' elements is systematically toward the telomere. Knowing that Y' elements are transcribed in strains deprived of telomerase activity [50], the co-orientation between the directionality of the BIR DNA synthesis and the Y' transcription is likely under strong selection pressure.

**ii. Heterochromatin.**   We observed increased ectopic BIR efficiency in the absence of Sir3 when BIR was initiated at 57 and 10 kb from the telomere. This suggests that BIR associated DNA synthesis is inhibited by heterochromatin, especially during the first and unstable phase of the reaction. These observations are in line with recent results showing a BIR inhibition by Sir2 when Sir2 is loaded on the BIR template [29]. In addition, subtelomeric heterochromatin prevents DNA end resection and therefore stabilizes the telomeric proximal DNA fragment and ensuing two-ended recombination [45]. Overall, heterochromatin promotes two-ended recombination at the expense of BIR in heterochromatic subtelomeric regions by directly impairing BIR, likely at the DNA synthesis step, and by impairing DNA end resection which prevents the loss of the telomere proximal fragment.

**iii. DNA modifying enzymes.** In addition to transcription, replication and chromatin that affect BIR in cis, different enzymes impair BIR. Exo1 and Sgs1 mediated DNA end resection impairs BIR both in a plasmid-based chromosome fragmentation assay and in a chromosomal assay [18,32,38,51]. The Mph1 helicase is also known to restrict BIR [29,39,40]. Interestingly, we found a higher BIR increase in the absence of Mph1 at telomere distal compared to telomere proximal regions. This may be explained by the longer exposure of the running D-loop to the action of Mph1. This running D-loop is also likely sensitive to structure specific nucleases able to cleave it since the combined absence of Mus81, Yen1 and Rad1 yields maximal BIR efficiency in an ectopic DSB repair chromosomal assay [39].

## Factors that promote ectopic BIR

BIR relies on the homologous recombination machinery, the polymerases delta, alpha and epsilon, and the Pif1 helicase. Factors protecting the long ssDNA generated behind the migrating D-loop also promote BIR. These factors include RPA, as well as Rad51 and Rad52, whose overexpressions increase BIR efficiency [18,38]. We recapitulated this latter property in our assay at the IVR-117 and IVR-57 BIR initiating loci but not at the IVR-10 locus, likely because of shorter ssDNA tracts not requiring additional Rad51 or Rad52 to be stabilized (Fig 5B). The Srs2 helicase is another factor that was shown to prevent toxic DNA interactions involving the long ssDNA generated behind the migrating D-loop [20]. Consistently, we found that Srs2 promotes BIR at any initiating locus tested in diploid cells. Unexpectedly, we observed this facilitation only when BIR is initiated 10 kb upstream of the telomere in a haploid background, while Elango et al. observed a BIR defect in the absence of Srs2 in a haploid background disomic for chromosome III [20]. This apparent discrepancy may come from different experimental conditions, such as the use of a chromosome fragmentation vector versus a chromosome III disome, or the use of a transformation-based assay involving asynchronous cells versus G2-arrested cells in which two copies of the template chromatid are present. Suspecting that the BIR defect we observed in the absence of Srs2 in haploid cells resulted from the presence of subtelomeric repeats, we further showed that Srs2 facilitates BIR independently of the LTR present in the IVR-10 BIR initiating locus, and independently of the subtelomeric Y' elements. The most likely hypothesis, still to be formally demonstrated, postulates that Srs2 promotes BIR in a haploid background close to telomeres by preventing ectopic recombination between the repeated X elements core sequences and / or between telomeric repeats themselves. Such a scenario would explain the defect in formation of survivors in the absence of telomerase observed in a *srs2* null mutant [52]. The link between the BIR defect in the absence of Srs2 and the presence of repeated sequences is further supported by the fact that the BIR defect in a diploid *srs2* null background is not suppressed by knocking out the *MATa* allele and therefore results from the ploidy of the cell. This diploid specific BIR phenotype observed in a *srs2* null background is in line with the diploid specific lethality of the *srs2K41A* helicase-dead mutant that accumulates toxic inter-homolog joint molecule intermediates [53]. It is also in line with the higher sensitivity to genotoxic agents of a *srs2* null diploid strain compared to a *srs2* null haploid strain [54]. Finally, the fact that this Srs2 phenotype is restricted to BIR events initiated in the vicinity of the telomere further supports the biphasic nature of the BIR-associated DNA synthesis step, with ssDNA being more accessible during the first phase of BIR.

In conclusion, multiple factors act in parallel to inhibit BIR and promote two-ended recombination when possible. Rad52, Rad59, RPA, Mph1 and MRX promote capture of the second end at two-ended DSBs which facilitates two-ended recombination at the expense of one-

ended recombination [29]. Here we show that ectopic BIR efficiency is further impaired by the extent of DNA to replicate, and by multiple chromosome features including transcription, heterochromatin and DNA repeats. In this context, it is important to note that even abortive BIR events promote genomic rearrangements like segmental duplications when coupled to joining with a chromosome fragment capped by a telomere [14,55–57]. Last but not least, it was recently shown that *S. cerevisiae* completes DNA replication after metaphase by a process reminiscent of mitotic DNA synthesis (MIDAS) observed in mammalian cells exposed to replication stress [58], and that this process primarily affects subtelomeric regions [59]. *S. cerevisiae* DNA replication therefore relies on an efficient MIDAS-like mechanism to replicate the last tens of kb of its chromosomes, which is precisely the range of size over which we found ectopic BIR to be the most efficient.

## Materials and methods

### Yeast strains and growth conditions

Yeast strains used in this study are derivatives of *S. cerevisiae* S288C and are listed in S1 Table. Standard media, growth conditions and genetic methods are as described in [60]. Genomic targeting experiments were performed by PCR-mediated gene replacement [61], followed by PCR analysis for discriminating correct and incorrect targeting. Details of the primers used for gene disruption and confirmation are available on request. The UV hyper-sensitivity of the *srs2* null mutants was verified. Translocations were constructed according to [35]. Briefly, two CRISPR-Cas9 mediated DSBs were generated at the two translocation breakpoints. Repair of these DSBs using two donor fragments containing the two translocation junctions yields the desired translocation. Induction of two DSBs is achieved by cloning two guide RNAs in the pGZ110 plasmid expressing Cas9. Donor fragments are 90 base pairs (bp) long DNA molecules containing two 45 bp regions flanking the translocation point. Translocations were confirmed by PCR and by pulse field gel electrophoresis (PFGE) as described in [22]. Oligonucleotides used for translocations are in S2 Table.

### Plasmids

Chromosome fragmentation vectors (CFVs) all derive from the CFV pLS192 described in [51]. All these plasmids contain a centromere, the *URA3* selection cassette, a telomere seed and a region of homology to a genomic locus to initiate BIR after linearization by SnaBI digestion (Fig 1). The plasmid pADW17 is identical to pLS192 but, in addition, contains an ARS (autonomously replicating sequence) to allow self-replication after transformation in yeast. The plasmid pADW17 is used as a control for yeast transformation efficiency.

CFVs were built by replacing the D8B genomic region containing *BUD3* from pLS192 by another genomic locus after EcoNI-BglII double digest [33,51]. Regions of interest were PCR amplified using a high-fidelity DNA polymerase generating blunt ended fragments, with one oligonucleotide containing either a BamHI or a BglII restriction site for subsequent semi-blunt cloning. The EcoNI site was made blunt after T4 DNA polymerase treatment. Plasmid pBL003 contains the *KANMX4* cassette. All other CFVs contain ~5 kb long genomic regions located on the right arm of chromosome IV. S3 Table lists all the CFVs used.

Plasmid pGZ110_synth4 was used as a substrate to generate novel combinations of pairs of guide RNAs cloned in the Cas9 expressing plasmid pGZ110 and carry out on-demand chromosomal translocations [35]. *RAD51* and *RAD52* overexpression was done using the high-copy-number plasmids pRS423_*RAD51* and pRS423_*RAD52*, respectively [18], and was checked by RT-qPCR (S4B Fig).

## BIR assay

Chromosome Fragmentation Vectors (CFVs) were digested with SnaBI for linearization and 100ng were used to transform competent yeast cells [32–34]. Cells were plated on synthetic medium lacking uracil to select for Ura+ transformants. In parallel, competent yeast cells were transformed with the circular CFV pADW17 plasmid to control for transformation efficiency. BIR efficiency corresponds to the number of transformants obtained with a linearized CFV divided by the number of transformants obtained with the circular pADW17 using the same amount of DNA. BIR efficiency of the *pif1m2* mutant was determined by using the pRS416 circular plasmid as a control. Indeed, pADW17 did not yield any transformant in this genetic background, likely because of the telomeric sequence it contains. Yeast transformants were counted after 3 days of growth at 30°C. BIR efficiencies were determined at least three times for each strain.

## Induction of transcription in the absence of thiamine

Thiamine-free medium was prepared using the Yeast Nitrogen Base Vitamin Free medium (FORMEDIUM), whose composition is based upon the formulation of the Yeast Nitrogen base except that all vitamins are omitted, and complemented with biotin 0.002 mg/l, Ca-panthotenic acid 0.4 mg/l, inositol 2 mg/l, p-aminobenzoic acid 0.2 mg/l. Thiamine 0.4 mg/l was added when required. Yeast cells were grown to stationary phase overnight with or without thiamine prior dilution the next morning in the same medium and grown to exponential phase. Yeast cells were transformed with the CFVs and transformants were selected on synthetic medium lacking uracil in the presence or absence of thiamine.

## RNA isolation and qPCR

Total RNA was isolated using the hot acid-phenol method [62], treated with RNase-free DNase I (Qiagen) and reverse transcribed using the Superscript III reverse transcriptase (Life technologies) with random hexanucleotide primers (Sigma-Aldrich). Quantitative PCR amplification of cDNA was carried out using the SYBR Premix Ex Taq II (Ozyme) using the following oligonucleotides: THI5_F (5'-AAGACTACACCGCCGTCA-3') and THI5_R (5'-AGCAAGCCAACTTGTCAATTC-3') for *THI5*; RAD51_cDNA_F (5'-TTCTTCCAC-CACGCGATTAG-3') and RAD51_cDNA_R (5'-AGATCGCGAACACACATTCA-3') for *RAD51*; RAD52_cDNA_F (5'-CTGCTAGCTCAAACCCAGAG-3') and RAD52_cDNA_R (5'-ACTCGCTGGAATATGCTTGG-3') for *RAD52*; ACT1_F (5'- CTATGT-TACGTCGCCTTGGA-3') and ACT1_R (5'-TTTGGTCAATACCGGCAGAT-3') for *ACT1* used as a control.

## Statistical analysis and modelling

All analyses were performed with the R environment (http://www.R-project.org/). The relationship between BIR efficiency and the length of DNA to synthesize (Fig 2B and S1 Fig) was fitted using the nonlinear least squares method implemented in R function nls(). The goodness of fit of the models was estimated using Akaike Information Criterion calculated with R function AIC(). Student's *t*-tests were performed using R function t.test() with the two-sided option. Differences in proportions were tested using Fisher's exact test for count data implemented in R function fisher.test().

## Supporting information

**S1 Fig. Fitting the relationship between BIR efficiency and the length of DNA to synthesize.** Combinations of two exponential functions are presented, with the first function fitting the data either from IVR-10 to IVR-26, or from IVR-10 to IVR-36, or from IVR-10 to IVR-41, or from IVR-10 to IVR-46.
(PDF)

**S2 Fig. BIR efficiency in *pol32* (panel A) and *pif1m2* (panel B) mutants.** Note that the pADW17 plasmid containing a telomeric seed was used as a transformation control for the *pol32* mutant but could not be used in the *pif1m2* mutants. Instead, the pRS416 plasmid was used. The transformation efficiency of pRS416 is higher than the transformation efficiency of pADW17. This likely explains the difference in BIR efficiency between the WT strains from the two graphs. Indicated are Student's *t*-test *p*-values <0.05 when comparing BIR efficiency from a given locus with its equivalent from the corresponding WT. WT values from the left graph are the same as in Fig 3 and represent the means of 49 independent experiments. The *pol32* values from the left graph are the means of three independent experiments. WT values from the right graph are the means of four independent experiments, and mutant values from the right graph are the means of three independent experiments. Error bars represent standard deviations.
(PDF)

**S3 Fig. PFGE analysis of BIR products.** Individual transformants were used to prepare DNA plugs that were analyzed by PFGE. Each gel lane corresponds to a unique transformant. Ethidium bromide stained gels (left) and corresponding radioactive signal from Southern blots using a *URA3* probe specific of the chromosome fragmentation vector (right) are shown. Arrowheads indicate chromosome fragments (CF) of abnormal size resulting from BIR initiated by the chromosome fragmentation vector. Gross chromosomal rearrangements (GCR) are indicated by arrows. **A.** Reference strain BY4741 transformed with the chromosome fragmentation vector initiating BIR at the IVR-10 locus. **B.** Reference strain BY4741 transformed with the chromosome fragmentation vector initiating BIR at the IVR-117 locus. **C.** Translocated IVR-113_XVI-12 strain transformed with the chromosome fragmentation vector initiating BIR at the IVR-117 locus. Note that chromosome IV from this strain is 101 kb shorter than its normal size. This is translated by a longer migration in the gel and a longer distance from chromosome XII as compared to all PFGE using non- translocated strains. **D.** *srs2* null strain transformed with the chromosome fragmentation vector initiating BIR at the IVR-10 locus. Note that GCRs involving chromosome XII that contains the rDNA locus were not considered. **E.** *srs2* null strain transformed with the circular chromosome fragmentation vector. Ethidium bromide stained gels only are shown since no BIR takes place in this context. The rationale of this experiment was to test for potential GCR in the absence of Srs2. No GCR were detected out of 100 clones.
(PDF)

**S4 Fig. Quantification of mRNA levels by RT-qPCR using *ACT1* for normalization. A.** Relates to Fig 4A. IVR-10_VIL-16 is the translocated strain containing *THI5* translocated downstream of the IVR-10 locus. Mean ratios of *THI5* and *ACT1* RT-qPCR quantifications from two independent experiments are indicated. Error bars indicate the deviation from the mean. **B.** Relates to Fig 5B. Ratios of *RAD51* and *RAD52* RT-qPCR quantifications normalized by the RT-qPCR quantification of *ACT1*. Overexpression of *RAD51* and *RAD52* was verified with one experiment only.
(PDF)

**S1 Table. List of strains.**
(DOCX)

**S2 Table. Oligonucleotides used to generate reciprocal translocations.**
(DOCX)

**S3 Table. Chromosome fragmentation vectors (CFVs) used.** All CFVs except pLS192 and pBL003 are designed to initiate BIR from chromosome IV regions, whose coordinates are from the *Saccharomyces* Genome Database (SGD). pLS192 is designed to initiate BIR from the left arm of chromosome III [51]. pBL003 contains the *KANMX4* cassette amplified from the BY4741 *rad27*::*KANMX4* strain using the U2 and D2 oligonucleotides [31] (http://www-sequence.stanford.edu/group/yeast_deletion_project/PCR_strategy.html). The resulting CFV is designed to initiate BIR from any *KANMX4* cassette whose transcription orientation is from the telomere to the centromere. BamHI (GGATCC) and BglII (AGATCT) sites are underlined. Note that the regions amplified to construct pBL013 and pBL014 contain a BamHI and a BglII site, respectively.
(DOCX)

**S1 Data. Numerical data supporting all graphs from the manuscript.**
(XLSX)

## Acknowledgments

We thank Mauro Modesti for support and ideas, Lance Langston and members of the Llorente lab for fruitful discussions.

## Author Contributions

**Conceptualization:** Tannia Uribe-Calvillo, Marie-Claude Marsolier, Bertrand Llorente.

**Formal analysis:** Tannia Uribe-Calvillo, Marie-Claude Marsolier, Bertrand Llorente.

**Funding acquisition:** Bertrand Llorente.

**Investigation:** Tannia Uribe-Calvillo, Laetitia Maestroni, Basheer Khadaroo, Christine Arbiol, Jonathan Schott.

**Methodology:** Tannia Uribe-Calvillo, Laetitia Maestroni, Marie-Claude Marsolier, Basheer Khadaroo, Christine Arbiol, Bertrand Llorente.

**Project administration:** Bertrand Llorente.

**Supervision:** Bertrand Llorente.

**Writing – original draft:** Tannia Uribe-Calvillo, Marie-Claude Marsolier, Bertrand Llorente.

**Writing – review & editing:** Marie-Claude Marsolier, Bertrand Llorente.

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
