## [Decision Letter · Decision Letter 0]

2 Apr 2022

Dear Dr Llorente, 

Thank you very much for submitting your Research Article entitled 'Comprehensive analysis of cis- and trans-acting factors affecting Break-Induced Replication' to PLOS Genetics.

The manuscript was fully evaluated at the editorial level and by independent peer reviewers. The reviewers appreciated the attention to an important topic but identified some concerns that we ask you address in a revised manuscript. It is especially important to  address the  following question  raised by the Reviewers 1 and 2: why allelic BIR can efficiently traverse 100 kb in allelic system but not in the transformation-based system used in your study? Together, we ask you to modify the manuscript according to the review recommendations. Your revisions should address the specific points made by each reviewer.

We hope to receive your revised manuscript within the next 60 days. If you anticipate any delay in its return, we would ask you to let us know the expected resubmission date by email to plosgenetics@plos.org.

[LINK]

Yours sincerely,

Anna Malkova

Guest Editor

PLOS Genetics

Gregory P. Copenhaver

Editor-in-Chief

PLOS Genetics

Reviewer's Responses to Questions

**Comments to the Authors:**

Reviewer #1: This new work demonstrates that break induced replication (BIR) is more efficient when strand invasion occurs closer to the telomere. This is shown using many new recombination assays on different chromosomes. A number of additional features and proteins dictates the efficiency of BIR like transcription, chromatin compaction, Mph1, the level of Rad51 and Rad52 and other. Altogether this is very elegant and well written study. After addressing questions listed below it should be considered for publication.

1. The major finding of this work is that BIR is more efficient when strand invasion occurs close to the telomere. What is not clear is whether the proximity to telomere and telomere specific features or only shorter DNA synthesis improve the efficiency of BIR?

2. The limitation of this work is that they look at ectopic BIR only (limited homology). With longer homology as in “allelic” BIR between truncated chromosome III and full size chromosome III as used by Malkova or Haber labs, BIR is nearly 100% efficient with the distance of over 100 kb to the telomere. It would be interesting for the authors to design one allelic assay and test whether BIR far away from chromosome ends increases when there is no homology limitation. Also the title and abstract should state that what is studied here is ectopic BIR and not BIR in general unless they study allelic BIR in a revised manuscript.

3. Rad51 and Rad52 overexpression needs to be documented by Western blots. It is important to know the fold increase in protein level. Alternatively, in case previous studies showed protein fold increase using the same multicopy plasmids, it would need to be stated.

4. Page 10 lane 1-2

“BIR events depend on POL32 and to a lesser extend on PIF1”

For this conclusion one would need to perform analysis in pif1delta not the hypomorphic mutant pif1-m2 that still partially localizes to nucleus. Therefore relative contribution of Pol32 and Pif1 to BIR cannot be estimated with present data. One set of experiments should be done in pif1delta or conclusion should be modified.

Minor

In discussion of the biphasic nature of BIR it is important to remember that BIR remains extremely mutagenic even 90 kb from strand invasion site, and strands are inherited in conservative manner likely all the way to the end of chromosome. This suggests that at least the basic mechanism of BIR is the same within the first 10-20 kb and 100 kb from strand invasion.

Reviewer #2: Llorente paper

This is a fine paper exploring the effect of template length and other factors on successful break-induced replication. The authors find that BIR copying the terminal 10-20 kb is much more successful than events that must copy a longer distance. Some of this has been inferred from a few data points previously, but the present MS is a careful analysis of the problem.

1. There is one intrinsic contradiction to this analysis that the authors should address. In a disome or diploid in which a truncated chromosome III is cleaved and repaired by BIR, the efficiencies of copying 100 kb are quite high (Malkova et al., 2005; Liu et al. 2021). Is this a feature of the fact that the broken chromosome is a pre-existing chromosome and not a fragment just introduced by transformation?

2. Two different normalizations are used in the paper: the repair of a BUD3-containing reference chromosome fragment or the co-transformation of a circular plasmid.

1. If the BUD3 is marked by the same URA3 as the test fragment, does this mean they come intrinsically from different experiments? Is this a problem, given the high variability of the outcomes in Fig. 2 (from 18% to 52%)? I can’t figure out how many different locations contribute to this big spread. Maybe they could be discriminated in some way (circles, squares, x, etc)?

Is BUD3 (100 kb from a telomere) an intrinsically low reference point? What’s the ratio between BUD3 repair and transformation with the circle?

2. What’s the marker on the reference circular plasmid?

3 p.6. data not shown… show it in a supplement. Are these data in the curve fitting?

4. The authors refer to repeated cycles of initiation, but isn’t this instability confined to the initial 20 or 30 kb? Later the authors suggest (as first suggested by Lydeard et al and then by Liu et al) that Pol� may take over later in the synthesis. This seems reasonable.

5. What to AIC numbers mean? How do they relate to confidence intervals?

6. p. 9 I am confused by this sentence: the probability of BIR reactions to be irreversibly

disrupted at each kb decreases after the synthesis of ~35-40 kb of DNA by a factor of

13, from 0.065 (1-0.935) to 0.005 (1-0.995). This required me to read it several times. Better:

“Over the first 35-40 kb, probability that BIR will be irreversibly disrupted is 0.065/kb , after which the probability falls by a factor of 13, to 0.005/kb.”

But, still, this seems hard to understand. If p(disruption) is 0.065/kb, then after 30 kb, is p = 1.95? So what am I missing? The decrease must not be a linear function. A careful description is needed.

7. Whether the effect of deleting Sir3 is on “accessibility” isn’t so clear to me. In MAT switching, where the donor is heterochromatic, there is very little effect on deleting sir3 on donor usage. Most relevant, Tasponina and Haber (2014) found that deleting Sir3 had no significant effect on template switching, which required a “jump” into the donor sequence (heterochromatic or not).

It might be useful to consult the Saccharomyces genome database to see the effect of sir mutations on the level of transcription of sub-telomeric genes.

8. p. 12 Dun1 does considerably more than regulate dNTP levels. Could its effect be related to changing the DNA damage response and to the levels of some HR or chromatin proteins?

9. p. 15 Interchromosomal jumps between solo delta elements was previously shown by Anand et al. (2014).

10. p. 17 define GCR

11. I am not clear how the authors’ view of Srs2 differs from that of Malkova’s lab? The effect of ploidy is curious. I wonder if BIR would be different in G2/M cells where there was a sister chromatid. (I am speculating here but I want to be clear that I have no expectation that the authors should do this analysis for the present paper!)

It may be relevant that Ira et al. (2002) found that srs2� decreased HO-induced ectopic recombination without crossovers but somehow had less effect on outcomes involving crossovers. Or not… very confusing protein.

Reviewer #3: The manuscript by Uribe-Calvillo et al describes a meticulous analysis of several of the properties that distinguish and define the BIR pathway. The authors made extensive use of cleverly designed BIR assays that included multiple engineered translocations to thoroughly characterize the already known negative relationship between BIR efficiency and length of the replication tract. The most valuable finding was the biphasic behaviour of the BIR efficiency plot where the authors were able to estimate a point where BIR likely transitions from short to long template extension modes, which possibly reflects a change in the replication machinery involved in the process. Overall, I thought this was a nice manuscript that will help move the BIR field forward in an appreciable way. The advancements reported in this work may not be huge or entirely novel, but the authors do provide important additional refinements on many of the key properties of BIR that had already been identified, but not explored in as much detail or in as much clarity as is provided here. Together, the various experiments reported should help better contextualize the instances where BIR is likely to happen in cells, relative to cases where chromosomal rearrangements (particularly non-reciprocal translocaitons) might look like BIR products, but are less likely to have actually arisen through BIR and thus were probably generated through cannonical 2-ended HR DSB repair.

I do not have any major concerns or suggestions that the authors would need to address.

**Have all data underlying the figures and results presented in the manuscript been provided?**

Reviewer #1: Yes

Reviewer #2: **No: **some data were listed as "not shown" and should be shown

Reviewer #3: Yes

PLOS authors have the option to publish the peer review history of their article (what does this mean?). If published, this will include your full peer review and any attached files.

Reviewer #1: No

Reviewer #2: **Yes: **James E. Haber

Reviewer #3: No

---

## [Decision Letter · Decision Letter 1]

19 May 2022

Dear Dr. Llorente,

We are pleased to inform you that your manuscript entitled "Comprehensive analysis of cis- and trans-acting factors affecting Break-Induced Replication" has been editorially accepted for publication in PLOS Genetics. Congratulations!

Before your submission can be formally accepted and sent to production you will need to complete our formatting changes, which you will receive in a follow up email. In addition, please consider the small correction that has been proposed by the Reviewer 2 which you can take care of as you prepare your final draft for the production team (the editorial team will not need to re-evaluate).

Please be aware that it may take several days for you to receive this email; during this time no action is required by you. Please note: the accept date on your published article will reflect the date of this provisional acceptance, but your manuscript will not be scheduled for publication until the required changes have been made.

Yours sincerely,

Anna Malkova

Guest Editor

PLOS Genetics

Gregory P. Copenhaver

Editor-in-Chief

PLOS Genetics

Comments from the reviewers (if applicable):

Reviewer's Responses to Questions

**Comments to the Authors:**

Reviewer #1: The revised manuscript is improved and suitable for publication.

Reviewer #2: The MS is much improved. One small correction: I added "over a distance of 100 kb"

Finally, it is intriguing that BIR between allelic loci from a disomic strain shows nearly

100% efficiency (Liu et al., 2021) over a distance of 100 kb, while between ectopic loci, such as in our

transformation-based assay and in chromosomal setups where BIR is initiated at an

HO-induced DSB (Davis and Symington, 2004; Donnianni and Symington, 2013;

Lydeard et al., 2007), BIR shows a lower efficiency and the length of DNA to replicate

negatively impacts BIR efficiency.

**Have all data underlying the figures and results presented in the manuscript been provided?**

Reviewer #1: Yes

Reviewer #2: Yes

PLOS authors have the option to publish the peer review history of their article (what does this mean?). If published, this will include your full peer review and any attached files.

Reviewer #1: No

Reviewer #2: **Yes: **James E. Haber

**Data Deposition**

http://datadryad.org/submit?journalID=pgenetics&manu=PGENETICS-D-22-00256R1

**Press Queries**

---

## [Editor Report · Acceptance letter]

15 Jun 2022

PGENETICS-D-22-00256R1 

Comprehensive analysis of cis- and trans-acting factors affecting ectopic Break-Induced Replication 

Dear Dr Llorente, 

We are pleased to inform you that your manuscript entitled "Comprehensive analysis of cis- and trans-acting factors affecting ectopic Break-Induced Replication" has been formally accepted for publication in PLOS Genetics! Your manuscript is now with our production department and you will be notified of the publication date in due course.

With kind regards,

Agnes Pap

PLOS Genetics

On behalf of:
